# Validation of automatically measured T1 map cortico-medullary difference (ΔT1) for eGFR and fibrosis assessment in allograft kidneys

Ibtisam Aslam[1,2], Fariha Aamir[2], Miklós Kassai[1], Lindsey A. Crowe[1], Pierre-Alexandre Poletti[1], Sophie de Seigneux[3], Solange Moll[4], Lena Berchtold[3], Jean-Paul Vallée[1]*

1 Service of Radiology, University Hospital of Geneva and Faculty of Medicine, University of Geneva, Geneva, Switzerland, 2 Medical Image Processing Research Group (MIPRG), Department of Electrical & Computer Engineering, COMSATS University Islamabad, Islamabad, Pakistan, 3 Service of Nephrology, Department of Medicine, University Hospital of Geneva, Geneva, Switzerland, 4 Department of Pathology, Institute of Clinical Pathology, University Hospital of Geneva, Geneva, Switzerland

* jean-paul.vallee@unige.ch

**Data Availability Statement:** All relevant data are within the article and its Supporting Information files.

## Abstract

MRI T1-mapping is an important non-invasive tool for renal diagnosis. Previous work shows that ΔT1 (cortex-medullary difference in T1) has significant correlation with interstitial fibrosis in chronic kidney disease (CKD) allograft patients. However, measuring cortico-medullary values by manually drawing ROIs over cortex and medulla (a gold standard method) is challenging, time-consuming, subjective and requires human training. Moreover, such subjective ROI placement may also affect the work reproducibility. This work proposes a deep learning-based 2D U-Net (RCM U-Net) to auto-segment the renal cortex and medulla of CKD allograft kidney T1 maps. Furthermore, this study presents a correlation of automatically measured ΔT1 values with eGFR and percentage fibrosis in allograft kidneys. Also, the RCM U-Net correlation results are compared with the manual ROI correlation analysis. The RCM U-Net has been trained and validated on T1 maps from 40 patients (n = 2400 augmented images) and tested on 10 patients (n = 600 augmented images). The RCM U-Net segmentation results are compared with the standard VGG16, VGG19, ResNet34 and ResNet50 networks with U-Net as backbone. For clinical validation of the RCM U-Net segmentation, another set of 114 allograft kidneys patient's cortex and medulla were automatically segmented to measure the ΔT1 values and correlated with eGFR and fibrosis. Overall, the RCM U-Net showed 50% less Mean Absolute Error (MAE), 16% better Dice Coefficient (DC) score and 12% improved results in terms of Sensitivity (SE) over conventional CNNs (i.e. VGG16, VGG19, ResNet34 and ResNet50) while the Specificity (SP) and Accuracy (ACC) did not show significant improvement (i.e. 0.5% improvement) for both cortex and medulla segmentation. For eGFR and fibrosis assessment, the proposed RCM U-Net correlation coefficient ($r$) and R-square ($R^2$) was better correlated ($r = -0.2$, $R^2 = 0.041$ with $p = 0.039$) to eGFR than manual ROI values ($r = -0.19$, $R^2 = 0.037$ with $p = 0.051$). Similarly, the proposed RCM U-Net had noticeably better $r$ and $R^2$ values ($r = 0.25$, $R^2 = 0.065$ with $p = 0.007$) for the correlation with the renal percentage fibrosis than the Manual ROI results ($r = 0.3$, $R^2 = 0.091$ and $p = 0.0013$). Using a linear mixed model, T1 was significantly higher

**Funding:** This work was supported by grants from the Clinical Research Center of the Medicine Faculty of Geneva University and Geneva University Hospital as well as the Louis- Jeantet Foundations and the Swiss National Science Foundation (JPV grant 32003B_159714, IZCOZO_177140 / 1 and SDS grant PP00P3_127454). This work was supported in part by the Centre for Biomedical Imaging (CIBM) and the Swiss National Science Foundation for its financial support for the PRISMA MRI (R'Equip grants: SNF No 326030_150816). The funders had no role in study design data collection and analysis decision to publish or preparation of the manuscript.

**Competing interests:** JPV, SDS have received grants from the Swiss Sciences Foundation, from the Geneva hospital university and the Geneva University. The MRI facility has been partially funded by the CIBM as well as a grant from Swiss National Science Foundation. This does not alter our adherence to PLOS ONE policies on sharing data and materials. The authors don't have any other financial or non-financial competing interest.

in the medulla than in the cortex ($p$<0.0001) and significantly lower in patients with cellular rejection when compared to both patients without rejection and those with humoral rejection ($p$<0.001). There was no significant difference in T1 between patients with and without humoral rejection ($p = 0.43$), nor between the types of T1 measurements (Gold standard manual versus automated RCM U-Net) ($p = 0.7$). The cortico-medullary area ratio measured by the RCM U-Net was significantly increased in case of cellular rejection by comparison to humoral rejection (1.6 +/- 0.39 versus 0.99 +/- 0.32, $p = 0.019$). In conclusion, the proposed RCM U-Net provides more robust auto-segmented cortex and medulla than the other standard CNNs allowing a good correlation of ΔT1 with eGFR and fibrosis as reported in literature as well as the differentiation of cellular and humoral transplant rejection. Therefore, the proposed approach is a promising alternative to the gold standard manual ROI method to measure T1 values without user interaction, which helps to reduce analysis time and improves reproducibility.

## Introduction

Chronic kidney disease (CKD) is a common condition referring to long-term kidney disease, defined as kidney function and/or structural changes for more than 3 months [1]. The estimated prevalence of CKD is 13.4% globally [2] and is a major public health issue. CKD is also the 16th most prominent reason of life lost worldwide [3].

Although initially asymptomatic, CKD can, if left undetected or untreated, lead to kidney failure which necessitates dialysis and eventually a kidney transplant. Progression of CKD to end-stage renal failure can be prevented by early detection and treatment [1, 3].

Interstitial fibrosis (IF) is one of the major predicting factors in CKD, independent of Estimated Glomerular Filtration Rate (eGFR) [4]. IF can currently only be assessed by kidney biopsy, an invasive examination which is difficult to perform repeatedly [1, 4].

Magnetic resonance imaging (MRI) is emerging as an important tool for non–invasive IF evaluation in the kidney [1, 4]. T1 mapping is a parametric map where each pixel of a kidney image represents the T1 spin-lattice relaxation time, which depends on the molecular environment [5]. In addition, the T1 is sensitive to pathological changes occurring in the tissues and can easily differentiate cortex and medulla. A recent study showed that the ΔT1 (cortico-medullary difference) has a significant correlation with fibrosis in CKD patients [4]. However, measuring the cortico-medullary values by manually drawing ROIs [1, 4, 6] on cortex and medulla (a gold standard method) is challenging, time-consuming and requires significant training. Moreover, manual-drawn ROIs may also affect reproducibility.

With the evolution of Artificial Intelligence (AI), Machine Learning (ML) and Deep Learning (DL), automatic medical image segmentation tasks are an active area of research [7]. Renal magnetic resonance imaging (Renal-MRI) has a great potential for segmentation of the kidney [8] and its associated compartments e.g. cortex, medulla etc. Several studies [8–13] have concentrated on manual and automatic segmentation of kidneys for specific renal cysts or total kidney volume using DL techniques.

T.L. Kline et al. proposed automatic semantic segmentation of kidney cysts in MR images of patients affected by kidney disease [9]. In this work, they developed and evaluated a fully automated semantic segmentation method to differentiate and analyse renal cysts in patients with autosomal-dominant polycystic kidney disease (ADPKD). They used an automated deep

learning U-Net [14] and ensemble technique for segmentation of renal cyst. U-Net is one of the popular used neural networks in deep learning. U-Net is an encoder-decoder network consists of multiple convolutional layers followed by activation layer, pooling layer, dropout, and batch normalization independent from encoding and decoding workflows.

K. Sharma et al. proposed a deep learning-based automated segmentation method for measuring total kidney volume (TKV) of ADPKD patients with mild to moderate or severe renal insufficiency [10]. The objective of this work was automated segmentation of ADPKD kidneys, and investigate its qualitative and quantitative accuracy and precision to measure TKV on a large dataset of CT acquisitions using fully convolutional neural networks (VGG16 [15]), trained end-to-end, on slices axial-CT sections. The automated method produced a fast and more reproducible measurement of the kidney volume as compared to manual measurements [10].

M. Haghighi et al. [16] proposed automatic whole kidney segmentation in dynamic contrast enhanced MRI (DCE-MRI) using a 3D Convolutional Neural Network (CNN). They proposed a time and memory efficient fully automated segmentation method which achieves high segmentation accuracy with running time in the order of seconds in both normal kidneys and hydronephrosis kidneys. They used two cascaded networks for the segmentation and localization of kidneys in both spatial and temporal information that performed well in both normal and abnormal kidneys [16].

A.J. Daniel et al. [11] also proposed automated renal segmentation in healthy controls (HCs) and CKD patients using deep learning 2D-CNN. In this study, they developed a fully automated method to segment T2-weighted MR images to calculate TKV of left and right kidneys. Results showed that 2D-CNN accurately segment the kidneys with higher accuracy than manual segmentation.

In all these reported studies, the performance of the Deep Learning (DL) methods for the medulla and cortex segmentation has not been evaluated. Therefore, this paper proposes a modified 2D U-Net to automatically segment the renal cortex and medulla (RCM U-Net) of T1 map in transplanted CKD patients to assess the renal fibrosis. The quality of the proposed RCM U-Net segmentation is compared with conventional DL methods i.e. VGG16 [15], VGG19 [17], ResNet34 [18], and ResNet50 [19] that all used U-Net as a backbone. The purpose of this work is to replace the gold standard manual ROI selection [1, 4, 6] over the cortex and medulla for estimation of cortico-medullary difference (ΔT1) with state of art automated segmentation approach. Moreover, this work also validated the automatically measured ΔT1 values estimated from auto-segmented cortex and medulla with gold standard manual ROI correlation results [4] for eGFR and fibrosis assessment of allograft kidneys. The eGFR was calculated according to the CKD-EPI equation [20].

## Material and methods

The proposed method involved pre-processing of the images and segmentation of cortex and medulla as shown in Fig 1.

### Pre-processing

The T1 map images are initially subjected to a pre-processing that is divided into 2 steps:

**Data augmentation.**   In DL, network training requires more data to train the network. To overcome this limitation, augmentation techniques [21] are applied to increase the amount of data while avoiding overfitting during training. In our experiment, initially data from 50 patients T1 maps (one central slice T1 map per patient) were manually segmented (i.e. full kidney, cortex and medulla) via 3D slicer [22] to make the labels for network training. From the manually segmented 3D slicer [22] data, 36 patients' images were used for network training, 4

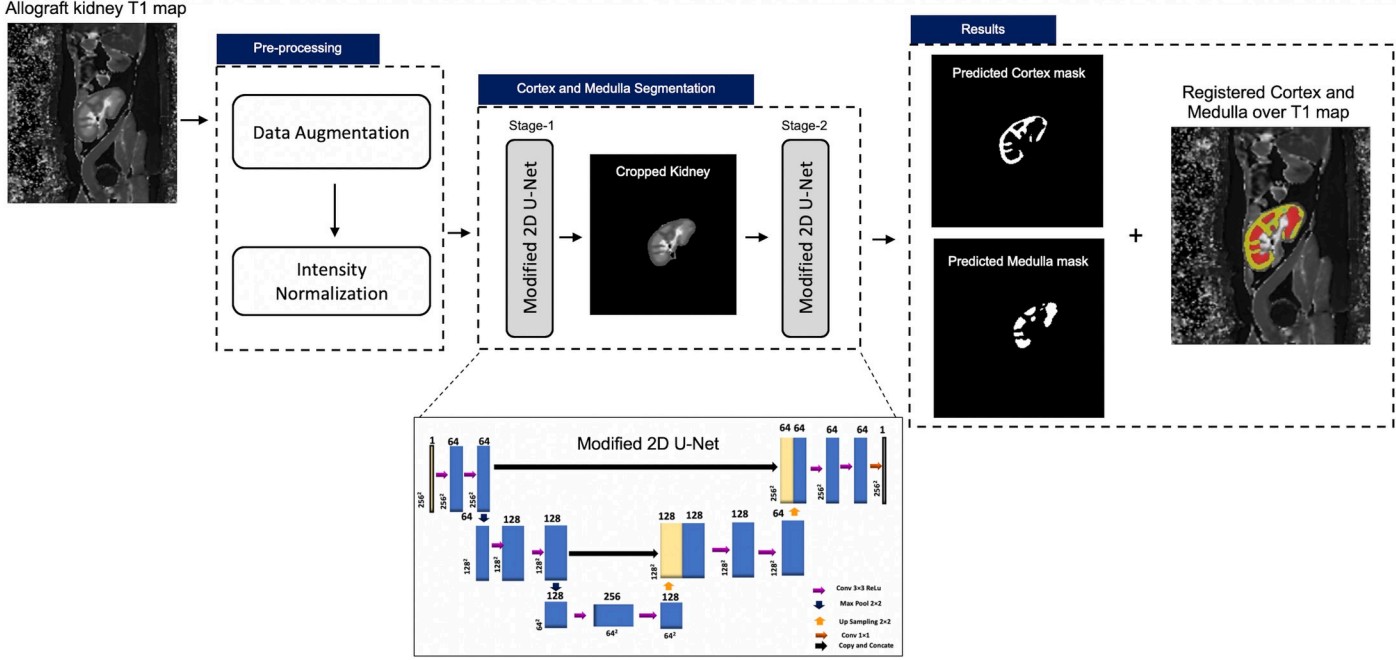

**Fig 1. Schematics diagram of proposed RCM U-Net to segment the cortex and medulla of transplanted kidneys.**

patients were used for validation while the other 10 patients' unseen images are used to test the trained network. Since, allograft kidneys could be in different shapes, sizes and transplanted in any orientation, data augmentation with affine transformation technique (e.g., rotation, flipping, shearing and translation etc.) [23] was used. After image augmentation, the 36 patients' data was augmented into 2160 training images to train the network and 4 patients' data was augmented into 240 validation images, while the 10 patients' unseen data was augmented into 600 test images to evaluate the network accuracy and robustness.

**Intensity normalization.** Intensity normalization [24] is also an important pre-processing step for medical image processing techniques such as image registration, segmentation, and reconstruction. During MR image acquisition, different conditions could lead to intensity variations between different subjects or the same subject at a different time even with the same parameters. This intensity variation will greatly undermine the performance of network. The kidneys can have very different intensities due to the its structure or compartments, i.e. cortex and medulla etc., which can hinder an eventual comparison of the texture characteristics of the T1 map. In this work, the intensity normalization [25] was done in MATLAB2021a® by using the following formula:

$$N = max\{0, min(I, 1)\}$$

Where, $I$ is the input image and $N$ is the normalized output.

## RCM U-Net

For the cortex and medulla segmentation, the RCM U-Net (at stage 1 & 2) was proposed as given in Fig 1.

In stage-1, the RCM U-Net was trained to segment the full kidney from the T1 map and remove the other same intensities background or other anatomy pixels that can complicate the true cortex and medulla segmentation (see Fig 1).

For stage-2, the stage-1 segmented kidneys were used to train the stage-2 RCM U-Net to segment cortex and medulla. Finally, after stage-2, the cortex and medulla segmentation results were predicted (as shown in Fig 1).

RCM U-Net consists of a symmetrical contraction and extraction path like the standard U-Net to capture the context and an accurate location of the allograft kidneys. In the proposed RCM U-Net, firstly two $3 \times 3$ convolution layers each followed by rectified linear unit (ReLU) as an activation function to solve the vanishing gradient problem are used. Secondly, we have applied a $2 \times 2$ max pooling operation with a stride of 2 for down-sampling. Max pooling helps to make the representation roughly invariant to limited translations of the input. In the decoder path, $2 \times 2$ up-sampling instead of max pooling was used to restore the output of original size. To get the desired size of the output image, up-sampling of the feature channels followed by a $3 \times 3$ convolution layer was performed. Also, a concatenation with the corresponding feature map from the contracting path with the expansion path was performed. In last layer, $1 \times 1$ convolution was used to combine each of the 64 features into one large feature map.

## Experimental setup and implementation

To train the proposed RCM U-Net architecture, from the 50 patients' data, a training set having data from 36 patients with a total of 2160 augmented images (i.e. rotation, flip, sheering etc.) and a validation set having data from 4 patients with a total of 240 images was used. The 240 images validation set optimize the internal losses during the network training. The trained network was tested on 10 patients unseen test data which was augmented to have 600 test images to evaluate the network performance and robustness.

To train the network, all the weights were initialized using a zero-centered normal distribution with a standard deviation of 0.05. In the proposed method, mean square error was used as a loss function which was minimized via RMSProp optimizer with a learning rate $1\times10^{-4}$ and weight decaying factor of 0.1. The proposed network training was implemented on Python 3.8 by Keras 2.3.1 using TensorFlow 2.1.0 [26] as backend on Intel(R) Xeon(R) CPU, 128 GB RAM, and GPU NVIDIA GeForce GTX 1080Ti, with a batch size of 5 for 200 epochs with patience = 50 as early stopping criteria. The quality of the proposed RCM U-Net segmentation was compared with standard VGG16 [15], VGG19 [17], ResNet34 [18], and ResNet50 [19] with U-Net as backbone which was trained on the same hardware. The parameters and hyper-parameters of each network were tuned after extensive experimentation.

The study was authorized by Geneva's local ethics committee for human studies (CER 11–160, Commission Cantonale d'Ethique de la Recherche) and carried out in accordance with the Helsinki Declaration principles. All patients gave written informed consent to participate in this study.

## Dataset

The proposed method was trained, validated and tested on 3000 augmented T1 map images from 50 allograft patients. The T1 map was acquired from a 3T Magnetom Prisma Fit MR scanner (Siemens AG, Erlangen, Germany) with the standard 32-element spine coil and the 18-element phased-array abdominal coil with the MOLLI T1 mapping sequence. The parameters were: resolution = $2\times2\times5mm^3$; TE/TR = 1.2/1500ms; iPAT = 2 (GRAPPA [27] reconstruction); flip angle = 35˚, inversion scheme 3(3)3(3)5, starting TI = 117ms, TI increment = 80ms.

For clinical validation of the proposed deep learning-based segmentation methods, another dataset of 114 kidney allograft patients (from the published study [4]) were auto-segmented to automatically measure cortical and medullary T1 as well as the cortico-medullary difference

**Table 1. Baseline characteristics of the study population (n = 114): Clinical parameters, medication, laboratory measurements, biopsy diagnosis and chronic histological lesions at the time of inclusion.**

| Characteristics | Total (n = 114) |
|---|---|
| **Clinical parameters** | |
| Age, years | 54 ± 13 |
| Male, n (%) | 76 (67%) |
| Body mass index, kg/m$^2$ | 25.1 ± 5.1 |
| Caucasian, n (%) | 105 (93%) |
| Living Donors | 53 (47%) |
| **Medication, n (%)** | |
| ACEi/ARB | 41 (36.3%) |
| Calcium channel blockers | 51 (45.1%) |
| Diuretics | 8 (7.1%) |
| Beta-blockers | 52 (46%) |
| Statins | 50 (44.3%) |
| Calcium supplementation | 66 (58.4%) |
| 1.25OH-vitamin D supplementation | 12 (10.6%) |
| 25OH-vitamin D supplementation | 89 (72.1%) |
| Anticalcineurin | 107 (94.7%) |
| Mycophenolate mofetil | 91 (80.5%) |
| Corticosteroids | 83 (74.1%) |
| **Laboratory measurements** | |
| Creatinine, micromol/l | 128 [99–148] |
| eGFR ml/min per 1.73m$^2$ * | 57 ± 20 |
| Hemoglobin, g/l (n = 187) | 130 ± 16 |
| Calcium, mmol/l (n = 154) | 2.39±0.11 |
| Phosphate, mmol/l (n = 165) | 0.97 ±0.20 |
| Magnesium, mmol/l | 0.67±0.1 |
| 25-hydroxyvitamin D, nmol/l | 77 ± 22 |
| Parathyroid hormone, pmol/l | 11 [6.0–13.0] |
| Albumin, g/l | 41 ± 3 |
| Urine protein/creatinin, g/24h | 0.51 [0.1–0.35] |
| Urine albumine/creatinin, mg/24h | 131 [13–64] |
| **Comorbidities** | |
| Hypertension | 104 (92%) |
| Diabetes | 27 (24%) |
| Smoking | 19 (18%) |
| Cardiovascular disease | 18 (16%) |
| Cancer | 4 (4%) |
| **Biopsy findings,** n (%) | |
| Cellular rejection | 4 (3.7%) |
| Humoral rejection | 9 (8.3%) |
| Tubular lesions | 22 (20.2%) |
| Glomerulonephritis | 18 (9.1%) |
| Vascular / FSGS | 2 (1.9%) |
| Chronic allograft nephropathy | 3 (2.8%) |
| Anticalcineurin toxicity | 39 (35.5%) |

(*Continued*)

**Table 1.** (Continued)

| Characteristics | Total (n = 114) |
|---|---|
| Fibrosis in % | 24% ± 14 |

Values reported as numbers and %, mean+/- SD or median with interquartile ranges, as appropriate.

*eGFR was calculated according to the CKD-EPI equation.

One biopsy may have more than one diagnosis.

ACEi/ARB, angiotensin-converting enzyme inhibitor/angiotensin II receptor blocker; FSGS, focal segmental glomerulosclerosis

ΔT1 values to assess the correlation with eGFR and Fibrosis as well as difference in case of kidney transplant rejection. These data were also acquired on a 3T Magnetom Prisma Fit MR scanner (Siemens AG, Erlangen, Germany) with the same parameters as given above. This study included adult kidney transplant recipients and CKD patients who were scheduled for a kidney biopsy for clinical grounds. MRI was performed on the same day as the biopsy wherever possible, or within one week. Patients over the age of 18 who were being monitored at our hospital were eligible for enrolment. Pregnancy, claustrophobia, and patient refusal were all exclusion criteria. Additional fasting serum and urine were collected and kept at -80˚C in all individuals. The Table 1 presents the baseline characteristics of the study population (n = 114): clinical parameters, medication, laboratory measurements, biopsy diagnosis and chronic histological lesions at the time of inclusion.

In this subgroup of patients, the manual ROIs drawn using Horos (https://horosproject.org/) [28] in the original publication [4] data were taken as gold standard to measure the mean cortex and medulla T1 values. The deep learning methods i.e. RCM U-Net (proposed method), VGG16 [15], VGG19 [17], ResNet34 [18], and ResNet50 [19], predicted the cortex and medulla masks which were used to extract the mean cortex and medulla T1 values. Additionally, we also performed ROI erosion for the deep learning-based cortex and medulla segmented masks to strongly match the hand-drawn gold standard ROIs. Furthermore, ROI erosion assisted in eliminating the overlapped area of cortex and medulla (segmentation leakage) in further analysis.

## Statistical analysis

The following evaluation metrics were used to assess the segmentations.

**Dice Coefficient (DC).** Dice Coefficient (DC) is a statistic that is used for the similarity comparison of two sets and known as the overlap index. DC measures the similarity level between the predicted results and the ground truths [29]. Mathematically DC is expressed as:

$$\text{Dice Coefficient (DC)} = \frac{2TP}{(2TP + FP + FN)} = \frac{2|X, Y|}{(|X| + |Y|)}$$

where, Y is the predicted mask and X is ground truth and TP is True Positive, FP is False Positive, and FN is False Negative.

**Accuracy (ACC).** Accuracy calculates how often predictions equal labels and it can be expressed mathematically as [30]:

$$\text{Accuracy} = \frac{(TP + TN)}{(TP + TN + FP + FN)}$$

where, TP is True Positive, TN is True Negative, FP is False Positive, and FN is False Negative.

**Sensitivity (SE).**   Sensitivity is defined as the actual proportion of positives that are correctly identified [29], and it can be expressed mathematically as [30]:

$$Sensitivity = \frac{TP}{TP + FN}$$

where, TP is True Positive, and FN is False Negative.

**Specificity (SP).**   Specificity is defined as the measure of the accurately identified proportion of actual negatives [29], and it can be expressed mathematically as [30]:

$$Specificity = \frac{TN}{TN + FP}$$

where, TN is True Negative, and FP is False Positive.

**Mean Absolute Error (MAE).**   Mean Absolute Error (MAE) is the amount of error in between the measured value (predicted) and true value (ground truth) [31]. It is mathematically expressed as:

$$MAE = \frac{\sum_{i=1}^{n} |Y_i - X_i|}{n}$$

where, $Y_i$ is the prediction and $X_i$ the true value.

**Percentage improvement.**   The percentage improvement is the amount of improvement between the final (proposed method) and initial (conventional method) values. It is mathematically expressed as:

$$\% \; improvement = \left(\frac{Y_i - X_i}{X_i}\right) \times 100$$

where, $Y_i$ is the final value and $X_i$ the initial value.

**Bland-Altman.**   The statistical significance between gold standard manual ROIs [4] and the deep learning based cortex, medulla and cortico-medullary difference values was tested using Bland-Altman (BA) [32] plots showing bias and 95% limit of agreement.

To study the utility of T1 in case of transplant rejection, a linear mixed model was used with T1 as the dependent variable using the R (Rstudio 2022.07.1 Build 554). The rejection status (cellular, humoral, no rejection), the measurement localization (cortex or medulla) and the type of measurement (Gold standard manual versus automated RCM U-Net) were fixed effects whereas the patient level was treated as a random effect. After checking for normality with a by Shapiro-Wilk's test, a linear mixed model with interactions was first computed to search for a significant effect of the fixed factors or their interactions. As no interactions was found, a second linear mixed model without interactions was used for the post-hoc Tukey's tests of the fixed effects. Finally, separated ANOVA for the cortex and medulla T1, the cortical and medullary area and the cortico-medullary area ratio were performed with the rejection status as a fixed effect and post-hoc Tukey's test.

## Results

The proposed RCM U-Net, as well as the other standard networks, were able to successfully segment, for the test images, the cortex and medulla on T1 maps in kidney allograft patients.

From visual assessment, the cortical and medullary segmentations seemed excellent for most of the networks. Fig 2 shows the standard 3D slicer (manual), automatically segmented cortex (green), medulla (red) and segmentation leakage (blue), a mixed portion of both cortex

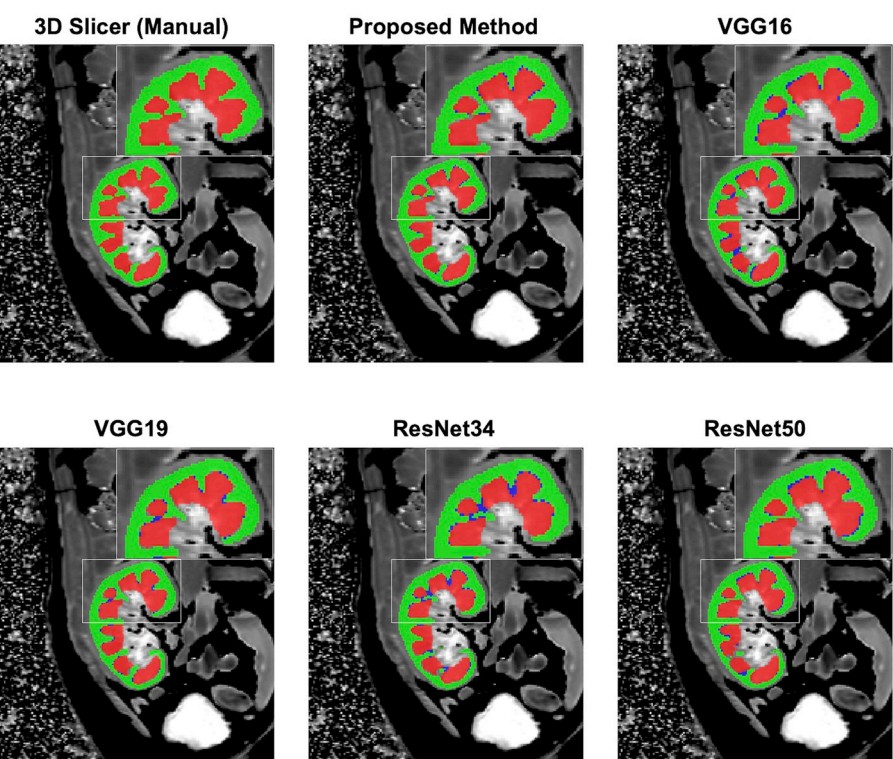

**Fig 2. Predicted cortex (Green)and medulla (Red) masks registered over reference T1 map.** Segmentation leakage (Blue) is a mixed portion of both cortex and medulla in segmentation provided by the all CNNs. The results show that the proposed RCM U-Net gives better segmentation results as compared to conventional CNNs i.e. VGG16 [15], VGG19 [17], ResNet34 [18], and ResNet50 [19] with U-Net as the backbone. Also, the conventional CNNs have more leakage segmentation than the proposed RCM U-Net as shown in zoomed image.

and medulla, registered over the reference T1 map. The proposed method showed better segmentation with less leakage (blue) for both cortex and medulla than the standard VGG16 [15], VGG19 [17], ResNet34 [18], and ResNet50 [19] as compared to standard 3D slicer manual segmentation.

Table 2 presents the performance evaluation metrics in terms of quantifying parameter i.e. DC, ACC, SP, SE and MAE for both cortex and medulla segmentation.

Fig 3 shows the mean percentage improvement of the proposed method with respect to VGG16 [15], VGG19 [17], ResNet34 [18], and ResNet50 [19] of cortex and medulla segmentation.

**Table 2. Evaluation metrics comparison of the proposed RCM U-Net, VGG16 [15], VGG19 [17], ResNet34 [18], and ResNet50 [19], for cortex and medulla segmentation.**

| Evaluation Metrics for Cortex | | | | | |
|---|---|---|---|---|---|
| Models | Dice Coefficient | Accuracy | Sensitivity | Specificity | Mean Absolute Error |
| Proposed Method | 0.918 | 0.993 | 0.946 | 0.999 | 0.007 |
| VGG16 | 0.871 | 0.982 | 0.897 | 0.998 | 0.018 |
| VGG19 | 0.879 | 0.982 | 0.907 | 0.998 | 0.018 |
| RESNET34 | 0.779 | 0.982 | 0.838 | 0.996 | 0.018 |
| RESNET50 | 0.764 | 0.982 | 0.816 | 0.997 | 0.018 |
| Evaluation Metrics for Medulla | | | | | |
| Models | Dice Coefficient | Accuracy | Sensitivity | Specificity | Mean Absolute Error |
| Proposed Method | 0.934 | 0.991 | 0.946 | 0.999 | 0.009 |
| VGG16 | 0.795 | 0.985 | 0.846 | 0.997 | 0.015 |
| VGG19 | 0.804 | 0.985 | 0.832 | 0.998 | 0.015 |
| RESNET34 | 0.757 | 0.985 | 0.846 | 0.997 | 0.015 |
| RESNET50 | 0.757 | 0.985 | 0.806 | 0.997 | 0.015 |

Comparing the scores in Table 2, the proposed method had the highest evaluation metrics values for both cortex and medulla segmentations as compared to conventional CNNs i.e. standard VGG16 [15], VGG19 [17], ResNet34 [18], and ResNet50 [19]. The proposed method had 60% and 40% less MAE (MAE = 0.007, 0.009) for cortex and medulla, respectively, than conventional CNNs (i.e. VGG16, VGG19, ResNet34 and ResNet50) while the SP and ACC did not show significant improvement. However, a promising trend was observed between proposed method and conventional CNNs [15, 17–19] results in terms of SE and DC score. The proposed method provided 16% & 17% and 20% & 23% better segmentation results (Fig 3) in terms of SE and DC (SE = 0.946, 0946; DC = 0.918, 0.934 –Table 2) as compared to the ResNet50 (SE:0.816, 0.806; DC = 0.764, 0.757) [19] while it showed 5% & 12 and 5% & 17% better segmentation results in terms of SE and DC than VGG16 (SE = 0.897, 0.846; DC = 0.871, 0.795 –Table 2) [15] for both cortex and medulla (S2 File).

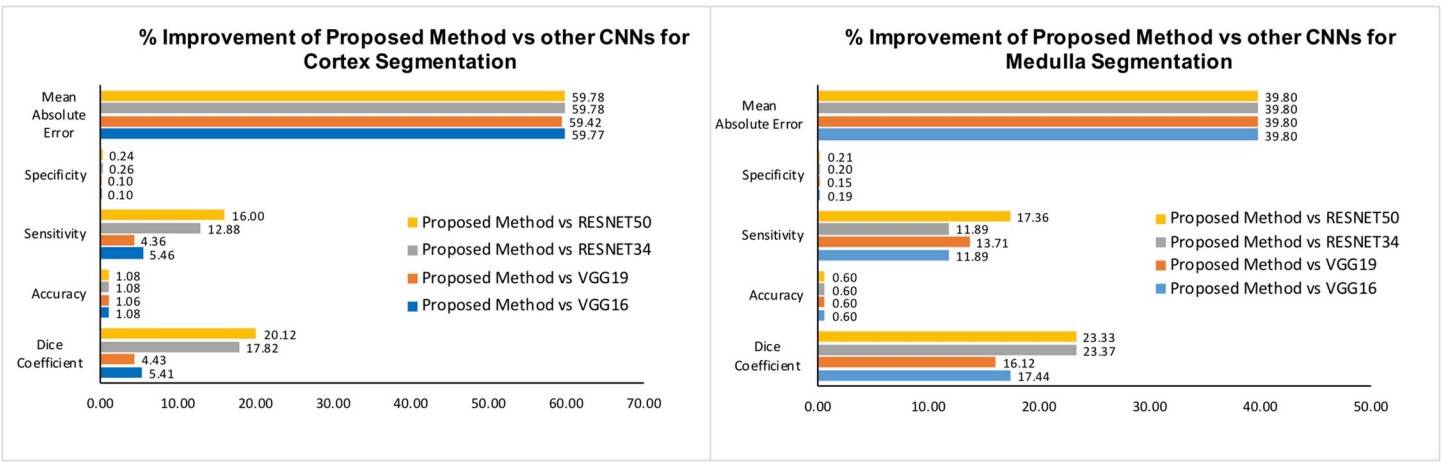

**Fig 3. Evaluation metrics in terms of percentage improvement of the proposed method (RCM U-Net), with respect to VGG16 [15], VGG19 [17], ResNet34 [18], and ResNet50 [19], of cortex and medulla segmentation.** These bar charts confirm the significant percentage improvement of the proposed method (RCM U-Net) over conventional CNNs.

## Validation of automatically measured ΔT1 values for eGFR and fibrosis assessment

The proposed RCM U-Net, VGG16 [15], VGG19 [17], ResNet34 [18], and ResNet50 [19] architectures were able to efficiently segment the cortex and medulla from the T1 maps in transplanted kidneys in all patients. Fig 4 shows the reference T1 map, gold standard manual ROIs and the deep-learning predicted masks registered on T1 maps from an allograft kidney patient. For the deep-learning predicted cortex and medulla masks, an ROI erosion was performed to strongly match the hand-drawn gold standard ROIs. Also, ROI erosion

**Reference T1 Map**

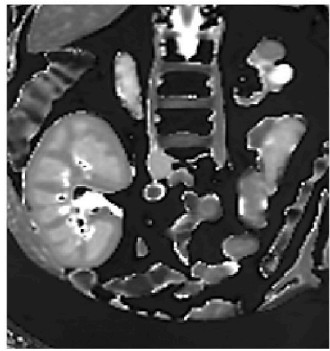

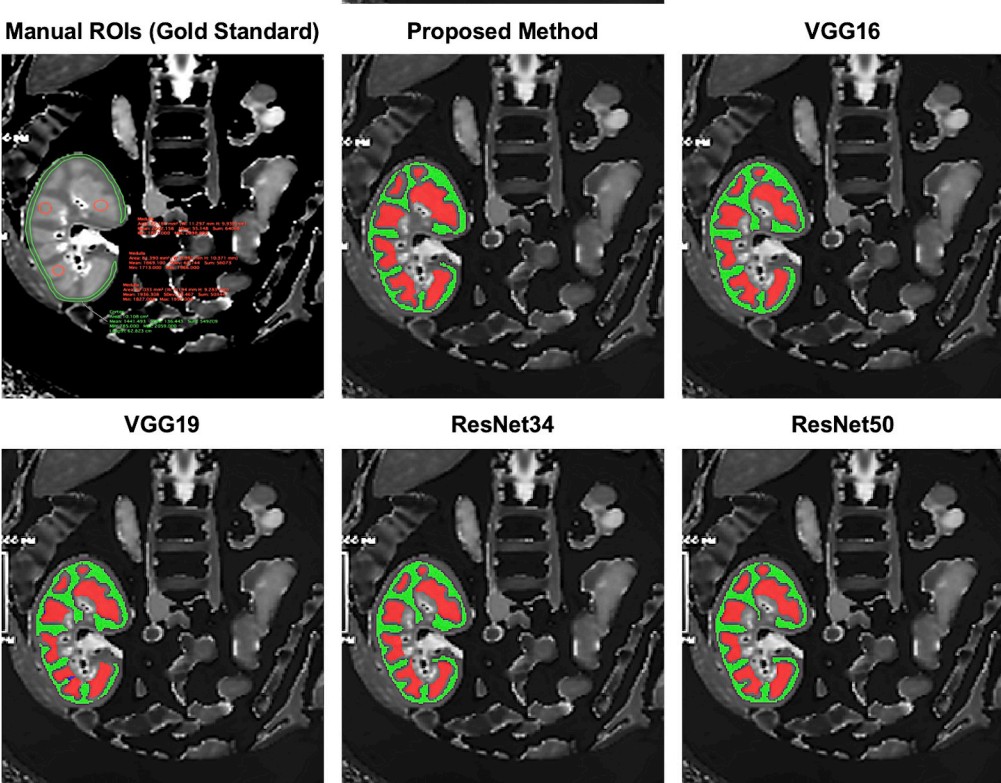

**Fig 4. The gold standard manually drawn ROIs (hand drawn ROIs around cortex and medulla in Horos) and the deep-learning predicted masks after erosion (Green: Cortex, Red: Medulla, Blue: Segmentation leakage) over an allograft kidney patient T1 map to compute the cortico-medullary values.** ROI erosion was performed to strongly match the hand-drawn gold standard ROIs. Also, ROI erosion was used to avoid the mixed portion of both cortex and medulla (segmentation leakage) to compute the cortico-Medullary values.

was used to avoid the mixed portion of both cortex and medulla (segmentation leakage) in the subsequent analysis.

As presented in Fig 5 and Table 3, the BA analysis showed good agreement between gold-standard manual ROI T1 values and automatically measured deep learning-based cortex, medulla T1 and cortico-medullary T1 difference values for all the networks. However, the proposed RCM U-Net demonstrated a better agreement with gold-standard manual ROIs values as compared to the other deep learning-based method i.e. VGG16 [15], VGG19 [17], ResNet34 [18], and ResNet50 [19] with a small and non-significant bias.

Figs 6 and 7 show correlations for T1 values vs eGFR and fibrosis (S3 File). The continuous line indicates least-square linear regression, while each dot represents a patient. The gold standard manual ROI correlation analysis published in [4] was reproduced and considered as reference for automatically measured ΔT1 correlation results. Both the proposed RCM U-Net and ResNet34 were able to give significant correlation for both the eGFR and renal fibrosis similar to the manual ROI method. In addition, the proposed RCM U-Net correlation coefficient ($r$) and R-square ($R^2$) results ($r$ = -0.2, $R^2$ = 0.041 with $p$ = 0.039) were closer to gold standard manual ROI correlation analysis [4] ($r$ = -0.19, $R^2$ = 0.037 with $p$ = 0.051) as compared to other networks [15, 17–19] for eGFR.

Similarly, from the correlation analysis between ΔT1 and fibrosis (shown in Fig 7), the deep learning-based methods showed a significant correlation with the gold standard manual ROI analysis [4]. The proposed RCM U-Net had better correlation results ($r$ = 0.25, $R^2$ = 0.065 with $p$ = 0.007) than the conventional VGG16 [15], VGG19 [17],

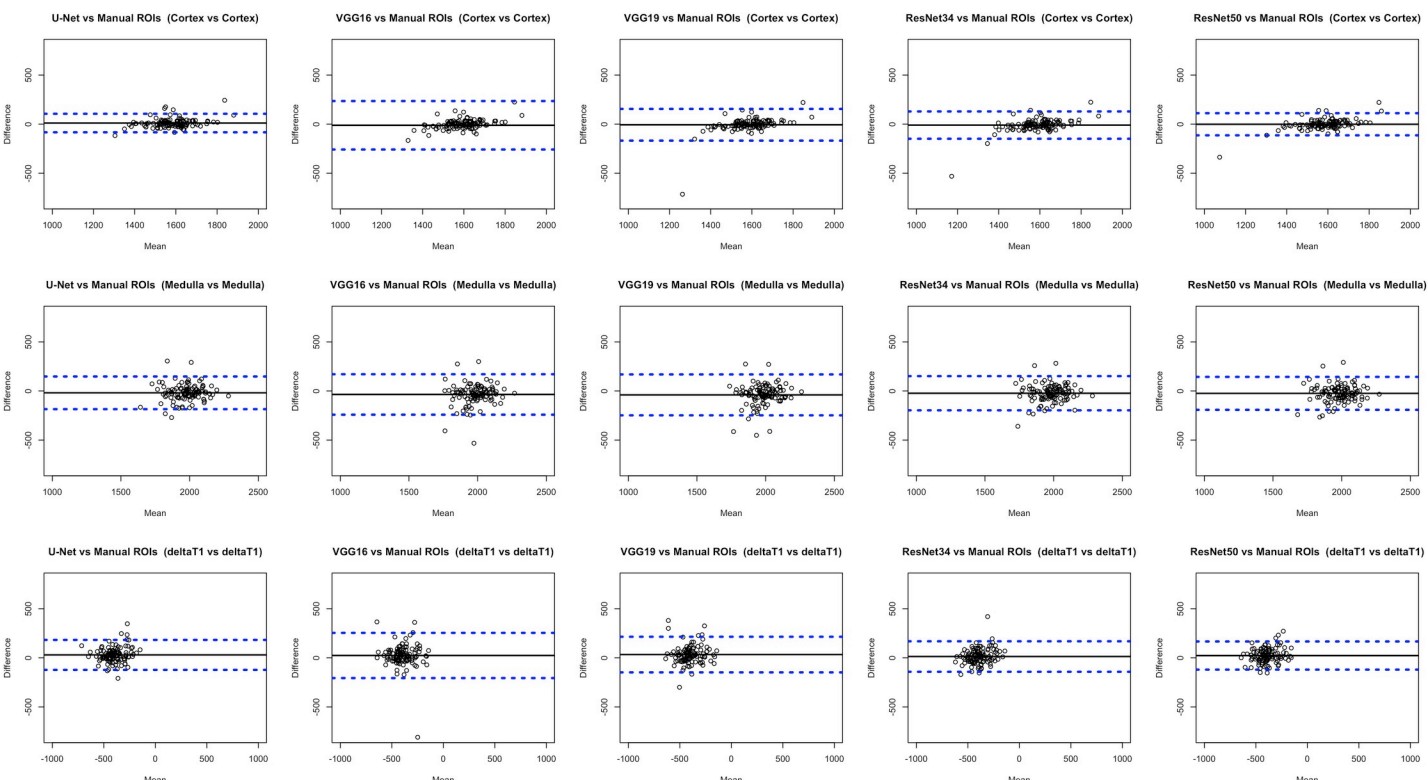

**Fig 5. Bland-Altman (BA) plots between gold standard manual ROIs [4] and deep learning based cortico-medullary values.** Top Row: BA plot of cortical T1 values; Middle Row: BA plot of medullary T1 values; Bottom Row: BA plot of cortico-medullary difference (ΔT1) values.

**Table 3. The BA plots statistical values for the gold standard manual ROIs [4] and the proposed RCM U-Net, [15], standard VGG16 [11], VGG19 [17], ResNet34 [18], and ResNet50 [19] automatically measured cortex, medulla and ΔT1 values.**

| Cortex | | | |
|---|---|---|---|
| **Models** | Mean Bias | Mean Lower LOA | Mean Upper LOA |
| **Proposed Method** | 10.97 | -83.40 | 105.35 |
| **VGG16** | -11.71 | -258.61 | 235.19 |
| **VGG19** | -6.57 | -168.21 | 155.06 |
| **RESNET34** | -9.88 | -149.06 | 129.30 |
| **RESNET50** | -1.17 | -113.98 | 111.64 |
| **Medulla** | | | |
| **Models** | Mean Bias | Mean Lower LOA | Mean Upper LOA |
| **Proposed Method** | -18.82 | -185.13 | 147.49 |
| **VGG16** | -35.69 | -241.78 | 170.41 |
| **VGG19** | -39.75 | -248.19 | 168.68 |
| **RESNET34** | -22.96 | -197.04 | 151.12 |
| **RESNET50** | -24.31 | -192.05 | 143.43 |
| **Delta T1** | | | |
| **Models** | Mean Bias | Mean Lower LOA | Mean Upper LOA |
| **Proposed Method** | 29.80 | -122.81 | 182.41 |
| **VGG16** | 23.98 | -206.25 | 254.22 |
| **VGG19** | 33.15 | -148.24 | 214.54 |
| **RESNET34** | 13.20 | -142.01 | 168.41 |
| **RESNET50** | 23.08 | -120.34 | 166.50 |

ResNet34 [18], and ResNet50 [19] with reference to the gold standard manual ROI [4] correlation ($r = 0.3$, $R^2 = 0.091$ and $p = 0.0013$).

From the linear mixed model analysis, the T1 was significantly higher in the medulla than in the cortex ($p < 0.0001$) and was significantly decreased in patients with cellular rejection by comparison to both patients without rejection and patients with humoral rejection ($p < 0.001$) as shown in Fig 8. There was no significant T1 difference between patients with a humoral rejection or without rejection ($p = 0.43$) as well as between the type of T1 measurements (Gold standard manual versus automated RCM U-Net) ($p = 0.7$). The ANOVA for both cortex and medulla showed that the medulla T1 was significantly decreased between patients with cellular or humoral rejections ($p = 0.026$). All the other T1 differences were not significant. The cortico-medullary area ratio was significantly increased in case of cellular rejection by comparison to humoral rejection (1.6 +/- 0.39 versus 0.99 +/- 0.32, $p = 0.019$).

## Discussion

This work proposes RCM U-Net for kidney cortex and medulla segmentation and validates the automatically measured T1 values of allograft kidneys for eGFR and fibrosis assessment and transplant rejection. With this approach, it was possible to segment cortex and medulla of transplanted kidneys and the proposed RCM U-Net performed better than other networks [15, 17–19] used in this work for both segmentation and clinical validation. The RCM U-Net automatically measured ΔT1 values showed a negative correlation with eGFR and a positive correlation with fibrosis as previously reported in literature [4]. The RCM U-Net also demonstrated a T1 difference in patients with kidney transplant rejection and patients with humoral rejection or absence of rejection.

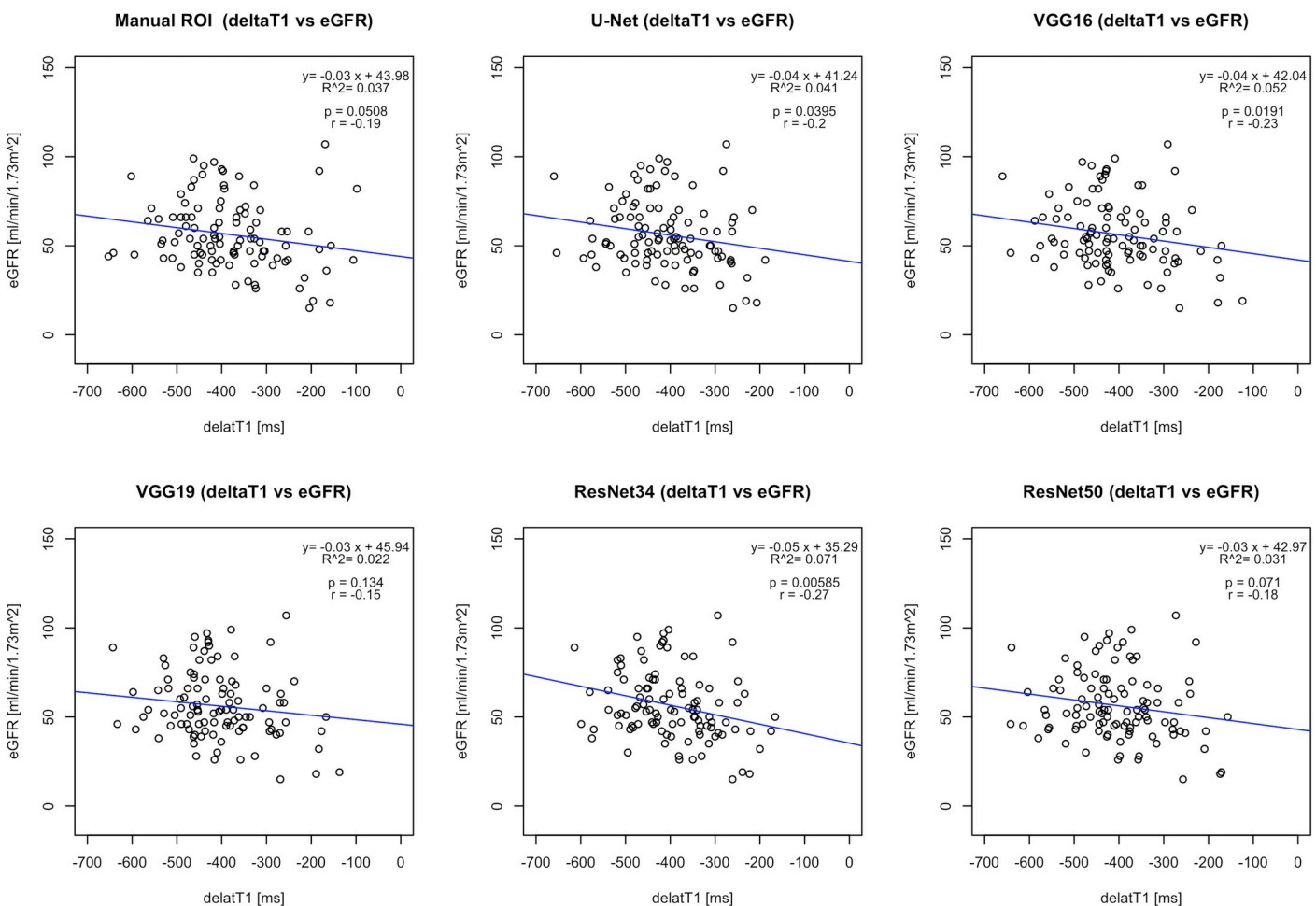

**Fig 6. The correlation analysis and scatter plots of gold-standard manual ROIs [4], deep learning-based methods ΔT1 values with eGFR.** The correlation results show negative correlation between MRI values and eGFR.

Previous studies have already successfully segmented renal images [8–13] but they were restricted to the whole kidney volume or the volume occupied by renal cysts. Our approach extends the whole kidney segmentation, which is the first step of our proposed RCM U-Net, to the segmentation of both cortex and medulla separately. Contrary to segmentation of renal cysts, that have a strong contrast with the rest of the renal parenchyma, differentiation of the cortex and medulla is much more difficult as signal intensities of both these regions are very close and tend to decrease with CKD and fibrosis. The accurate segmentation of both cortex and medulla with RCM U-Net is important especially since it has been obtained in a CKD cohort [4].

One additional originality of our work is a physiological validation of our proposed method, where most of the other segmentation work used only standard image processing metrics [14, 18, 29] to evaluate the network performance. With our automated measurements, we were able to obtain similar correlation results as reported in [4] for both eGFR or renal percentage fibrosis. This is an important achievement as the cortico-medullary difference of various MRI parameters seems to be a more powerful indicator of renal diseases or evolution than the individual absolute values. This has been well demonstrated with the diffusion weighted MRI (DWI) where ΔADC (apparent diffusion coefficient) is better correlated to both renal

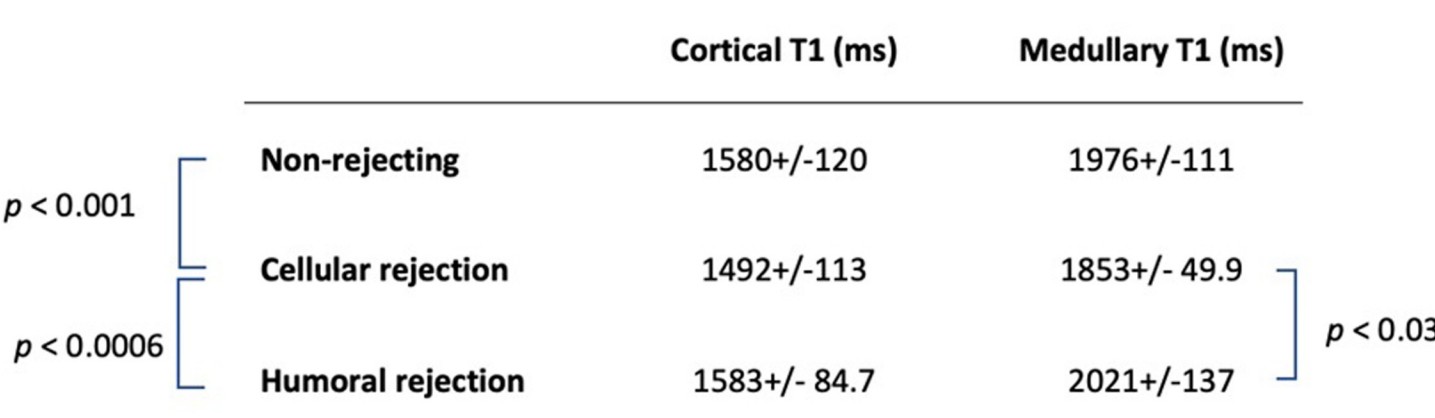

**Fig 7. The correlation analysis and scatter plots of gold-standard manual ROIs [4], deep learning-based methods ΔT1 values with percentage fibrosis.** The correlation results show postive correlation between MRI values and percentage fibrosis.

|  | Cortical T1 (ms) | Medullary T1 (ms) |
|---|---|---|
| **Non-rejecting** | 1580+/-120 | 1976+/-111 |
| **Cellular rejection** | 1492+/-113 | 1853+/- 49.9 |
| **Humoral rejection** | 1583+/- 84.7 | 2021+/-137 |

$p < 0.001$ (between Non-rejecting and Cellular rejection)

$p < 0.0006$ (between Cellular rejection and Humoral rejection)

$p < 0.03$ (between Cellular rejection and Humoral rejection, Medullary T1)

**Fig 8. T1 values (mean +/- standard deviation in ms) measured by the automated RCM U-Net method in the cortex and medulla of transplant patients according to their status (no rejection, cellular or humoral rejection) with the corresponding significant p values from the linear mixed models' analysis.**

fibrosis or renal failure evolution [4] than either cortical or medullary ADC alone. Therefore, our method should promote the use of systematic quantification of cortex and medulla and can be extended to ADC in further renal studies. Our proposed deep learning-based methods could replace the gold standard manual ROI method to measure cortico-medullary values; this will help to reduce the burden in terms of time to manually draw ROIs for cortex and medulla and will help to improve reproducibility. Furthermore, our automated method was able to provide unsupervised T1 measurements of both the renal cortex and medulla in the range of previously published values using a similar MOLLI (MR) sequence and magnetic field (3T) [33]. As a demonstration of our method, we studied the T1 difference between rejecting and non-rejecting transplants in our patient's group. T1 was decreased in patients with cellular rejection but not with humoral rejection or non-rejecting transplant using a linear mixed model integrating all the available T1 measurements. This T1 difference between the cellular and humoral rejections was significant in the medulla but not the cortex when separated analysis were performed. The absence of T1 difference in the cortex may result from the small number of patients with transplant rejection in our study (n = 13). The origin of the T1 decrease in patients with cellular rejection is currently not known as T1 measurements are sensitive to both the interstitial space and the intracellular compartment. Fibrosis is not likely to be involved. Although, fibrosis is positively correlated with T1 in the cortex [6], it was not significantly different between the rejection groups. A modification of the intracellular space secondary to the inflammatory cell infiltrate may be an explanation as T1 is sensitive to cellular density and type as shown recently in an experimental tumor model [34]. This is further supported by the observation of an increase cortico-medullary area ratio in patients with cellular rejection. The cortico-medullary area ratio is a new index provided directly by our user independent method that may improve the assessment of the cortical thickness and cortico-medullary difference. Further studies are needed to confirm these preliminary results.

We observed an overall better performance of our proposed RCM U-Net over the other networks used in this work. All the networks were trained, validated and tested on same data, with same hardware and tuned to their best parameters and hyperparameters after extensive experimentation. The proposed RCM U-Net has symmetrical encoder-decoder and recovers the data features efficiently with better segmentation results. By comparison, the VGG16 [15], VGG19 [17], ResNet34 [18], and ResNet50 [19] based encoders with U-Net backbone have asymmetrical encoder-decoder layers which could cause some information loss in the expansion path.

Despite the promising results and advantages, our study has some limitations. The gold standard for the training is a manually drawn ROI that can be subject to errors especially in the case of poor image quality. However, it remains the best available method as there is currently no other strategies to isolate in a non-invasive way the medulla and cortex. To train the network from scratch, needed much intensive work of manual kidney and compartment segmentation due to the supervised learning problem. However, once the network is fully trained, it can segment the cortex and medulla within seconds. Secondly, to correlate the automatically measured ΔT1 values with gold standard manual ROIs [4], the auto-segmented cortex and medulla from proposed network, requires ROI erosion to strongly mimic the hand-drawn ROIs. Also, this work uses only 50 manually segmented transplanted kidneys and compartments (cortex and medulla), followed by data augmentation to train the network. To improve the network performance and robustness, more manually cropped data followed by multiple data augmentation transforms could be used. So far, the proposed network has been applied to a single type of images (i.e. the renal T1 map). It remains to be determined if it performs so well with other contrast (such as T2, T2* or DWI). Finally, our method has only been applied to transplanted kidneys, which are less susceptible to respiratory motion than native kidney.

Therefore, its performance in the segmentation of native kidney is not known but it will be the subject of further studies.

## Conclusion

This paper proposes RCM U-Net based automated cortex and medulla segmentation for T1 map images of transplanted kidneys in CKD and validates the automatically measured ΔT1 values with the gold standard manual ROI correlation results for eGFR and fibrosis assessment. It also showed its potential to assess kidney rejection. The proposed method could be an alternative to the gold standard manual ROI method, subjective ROI placement, to measure the ΔT1 values which helps to reduce the human effort and will improve reproducibility.

## Supporting information

**S1 File. List of list of acronyms/abbreviations used in the paper.**
(DOCX)

**S2 File. AI network evaluation matrices.**
(XLSX)

**S3 File. eGFR & fibrosis vs imaging values.**
(XLSX)

## Author Contributions

**Conceptualization:** Ibtisam Aslam.

**Data curation:** Miklós Kassai.

**Formal analysis:** Ibtisam Aslam, Lindsey A. Crowe, Jean-Paul Vallée.

**Funding acquisition:** Jean-Paul Vallée.

**Investigation:** Ibtisam Aslam, Miklós Kassai, Lindsey A. Crowe.

**Methodology:** Ibtisam Aslam, Fariha Aamir.

**Project administration:** Jean-Paul Vallée.

**Resources:** Lindsey A. Crowe, Jean-Paul Vallée.

**Software:** Ibtisam Aslam.

**Supervision:** Lindsey A. Crowe, Jean-Paul Vallée.

**Validation:** Ibtisam Aslam.

**Visualization:** Miklós Kassai, Jean-Paul Vallée.

**Writing – original draft:** Ibtisam Aslam.

**Writing – review & editing:** Miklós Kassai, Lindsey A. Crowe, Pierre-Alexandre Poletti, Sophie de Seigneux, Solange Moll, Lena Berchtold, Jean-Paul Vallée.

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
