## [Decision Letter · Decision Letter 0]

4 Aug 2022

PONE-D-22-08551

Validation of automatically measured T1 map cortico-medullary difference (ΔT1) for eGFR and Fibrosis assessment in allograft kidneys

PLOS ONE

Dear Dr. Vallée,

Thank you for submitting your manuscript to PLOS ONE. After careful consideration, we feel that it has merit but does not fully meet PLOS ONE’s publication criteria as it currently stands. Therefore, we invite you to submit a revised version of the manuscript that addresses the points raised during the review process.

Two external reviewers have evaluated your manuscript, and have raised some concerns about the presentation of the manuscript and clarity of the methods. When preparing your revisions, please pay particular attention to addressing Reviewer 2's queries regarding the approach to patient recruitment and patient demographics. 

A marked-up copy of your manuscript that highlights changes made to the original version. You should upload this as a separate file labeled 'Revised Manuscript with Track Changes'.An unmarked version of your revised paper without tracked changes. You should upload this as a separate file labeled 'Manuscript'.

We look forward to receiving your revised manuscript.

Kind regards,

Jamie Males

Editorial Office

PLOS ONE

Journal Requirements:

"This work was supported by grants from the Clinical Research Center of the Medicine Faculty of Geneva University and Geneva University Hospital, as well as the Leenaards and Louis- Jeantet Foundations and the Swiss National Foundation (JPV grant 320038_159714, IZCOZO_177140 / 1 and SDS grant PP00P3_127454). This work was supported in part by the Centre for Biomedical Imaging (CIBM) of EPFL, University of Geneva and the University Hospitals of Geneva and Lausanne and the Swiss National Foundation for its financial support for the PRISMA MRI (R’Equip grants: SNF No 326030_150816)"

"This work was supported by grants from the Clinical Research Center of the Medicine Faculty of Geneva University and Geneva University Hospital, as well as the Leenaards and Louis- Jeantet Foundations and the Swiss National Foundation (JPV grant 320038_159714, IZCOZO_177140 / 1 and SDS grant PP00P3_127454). This work was supported in part by the Centre for Biomedical Imaging (CIBM) of EPFL, University of Geneva and the University Hospitals of Geneva and Lausanne and the Swiss National Foundation for its financial support for the PRISMA MRI (R’Equip grants: SNF No 326030_150816)"

"Additionally, because some of your funding information pertains to [commercial funding//patents], we ask you to provide an updated Competing Interests statement, declaring all sources of commercial funding. 

In your Competing Interests statement, please confirm that your commercial funding does not alter your adherence to PLOS ONE Editorial policies and criteria by including the following statement: "This does not alter our adherence to PLOS ONE policies on sharing data and materials.” as detailed online in our guide for authors  http://journals.plos.org/plosone/s/competing-interests.  If this statement is not true and your adherence to PLOS policies on sharing data and materials is altered, please explain how"

Please include the updated Competing Interests Statement and Funding Statement in your cover letter. We will change the online submission form on your behalf.

Reviewers' comments:

Reviewer's Responses to Questions

**Comments to the Author**

1. Is the manuscript technically sound, and do the data support the conclusions?

Reviewer #1: Yes

Reviewer #2: Yes

2. Has the statistical analysis been performed appropriately and rigorously? 

Reviewer #1: Yes

Reviewer #2: Yes

3. Have the authors made all data underlying the findings in their manuscript fully available?

Reviewer #1: Yes

Reviewer #2: Yes

4. Is the manuscript presented in an intelligible fashion and written in standard English?

Reviewer #1: Yes

Reviewer #2: Yes

5. Review Comments to the Author

Reviewer #1: Dear Author/s

The article seems interesting in terms of content and preparation. The rate of chronic kidney failure is increasing with the increase of comorbid diseases (hypertension, diabetes, etc.) worldwide. As a renal replacement therapy, kidney transplantation is superior to other methods (hemodialysis, peritoneal dialysis).

The patient's primary disease, treatment compliance, immunosuppression, etc. are important in the success of the transplant. It is used in biopsy and imaging as well as laboratory in follow-ups. Scanning of the cortex medulla in MRI, as stated in the study, contributes to the diagnosis of interstitial fibrosis. In addition, confirmation of this in the laboratory and the contribution of image data in patients who develop rejection will help the clinician.

-One of the most important shortcomings of the study is the inability to compare clinical data with imaging. Is there a relationship between renal function, rejection and fibrosis with imaging?

The work will be more valuable if the specified features are highlighted.

Best regards

Reviewer #2: Overall this is a welcome paper, which should move the field of MRI kidney onwards. Current methods for defining ROIs are time consuming and automated methods are attractive. Overall the paper looks technically sounds and the methods are appropriate . I have a few suggestions

1. there are a huge number of acronyms. I appreciate the challenges with this work but at no point could I find u-NET described and as a clinician this was very confusing. please describe what this is. Likewise many other values are presented as a series of letters which limited the understanding of this paper

2. please supply some clinical information about how patients recruited etc

3. some baseline demographics would be helpful

4. what eGFR formula was used

5. I could not find a figure legend- maybe it was lost in formatting

6. typo in figure 3

6. PLOS authors have the option to publish the peer review history of their article (what does this mean?). If published, this will include your full peer review and any attached files.

Reviewer #1: **Yes: **Yavuz AYAR

Reviewer #2: No

---

## [Author Response · Author response to Decision Letter 0]

20 Sep 2022

Response to Reviewers

Authors thank the all reviewers to provide such valuable comments and feedback to make this paper clearer and more concise. The black color text is the response to the reviewer’s comments while the red color text is included into the manuscript. 

Reviewer: 1

We appreciate reviewer 1 comments. The response to the Reviewer 1 comments are given below:

1. The article seems interesting in terms of content and preparation. The rate of chronic kidney failure is increasing with the increase of comorbid diseases (hypertension, diabetes, etc.) worldwide. As a renal replacement therapy, kidney transplantation is superior to other methods (hemodialysis, peritoneal dialysis).

The patient's primary disease, treatment compliance, immunosuppression, etc. are important in the success of the transplant. It is used in biopsy and imaging as well as laboratory in follow-ups. Scanning of the cortex medulla in MRI, as stated in the study, contributes to the diagnosis of interstitial fibrosis. In addition, confirmation of this in the laboratory and the contribution of image data in patients who develop rejection will help the clinician.

One of the most important shortcomings of the study is the inability to compare clinical data with imaging. Is there a relationship between renal function, rejection and fibrosis with imaging? The work will be more valuable if the specified features are highlighted.

Answer: To study the utility of T1 in case of transplant rejection, a linear mixed model was used with T1 as the dependent variable using the R (Rstudio 2022.07.1 Build 554). The rejection status (cellular, humoral, no rejection), the measurement localization (cortex or medulla) and the type of measurement (Gold standard manual versus automated RCM U-Net) were fixed effects whereas the patient level was treated as a random effect. After checking for normality with a by Shapiro-Wilk’s test, a linear mixed model with interactions was first computed to search for a significant effect of the fixed factors or their interactions. As no interactions was found, a second linear mixed model without interactions was used for the post-hoc Tukey’s tests of the fixed effects. Finally, separated ANOVA for the cortex and medulla T1, the cortical and medullary area and the cortico-medullary area ratio were performed with the rejection status as a fixed effect and post-hoc Tukey’s test (Added in revised manuscript Line: 282-292).

From the linear mixed model analysis, the T1 was significantly higher in the medulla than in the cortex (p < 0.0001) and was significantly decreased in patients with cellular rejection by comparison to both patients without rejection and patients with humoral rejection (p < 0.001) as shown in Table 4. There was no significant T1 difference between patients with a humoral rejection or without rejection (p = 0.43) as well as between the type of T1 measurements (Gold standard manual versus automated RCM U-Net) (p = 0.7). The ANOVA for both cortex and medulla showed that the medulla T1 was significantly decreased between patients with cellular or humoral rejections (p = 0.026). All the other T1 differences were not significant. The cortico-medullary area ratio was significantly increased in case of cellular rejection by comparison to humoral rejection (1.6 +/- 0.39 versus 0.99 +/- 0.32, p = 0.019). (Added in revised manuscript Line: 426-435).

Table 4: T1 values (mean +/- standard deviation in ms) measured by the automated RCM U-Net method in the cortex and medulla of transplant patients according to their status (no rejection, cellular or humoral rejection) with the corresponding significant p values from the linear mixed models’ analysis (Added in revised manuscript Line: 436-440).

.

Furthermore, our automated method was able to provide unsupervised T1 measurements of both the renal cortex and medulla in the range of previously published values using a similar MOLLI (MR) sequence and magnetic field (3T) [35]. As a demonstration of our method, we studied the T1 difference between rejecting and non-rejecting transplants in our patient’s group. T1 was decreased in patients with cellular rejection but not with humoral rejection or non-rejecting transplant using a linear mixed model integrating all the available T1 measurements. This T1 difference between the cellular and humoral rejections was significant in the medulla but not the cortex when separated analysis were performed. The absence of T1 difference in the cortex may result from the small number of patients with transplant rejection in our study (n=13). The origin of the T1 decrease in patients with cellular rejection is currently not known as T1 measurements are sensitive to both the interstitial space and the intracellular compartment. Fibrosis is not likely to be involved. Although, fibrosis is positively correlated with T1 in the cortex [6], it was not significantly different between the rejection groups. A modification of the intracellular space secondary to the inflammatory cell infiltrate may be an explanation as T1 is sensitive to cellular density and type as shown recently in an experimental tumor model [36]. This is further supported by the observation of an increase cortico-medullary area ratio in patients with cellular rejection. The cortico-medullary area ratio is a new index provided directly by our user independent method that may improve the assessment of the cortical thickness and cortico-medullary difference. Further studies are needed to confirm these preliminary results (Added in revised manuscript Line: 473-492).

Also, abstract is updated in revised manuscript from line: 43-50 as following:

Using a linear mixed model, T1 was significantly higher in the medulla than in the cortex (p<0.0001) and significantly lower in patients with cellular rejection when compared to both patients without rejection and those with humoral rejection (p<0.001). There was no significant difference in T1 between patients with and without humoral rejection (p=0.43), nor between the types of T1 measurements (Gold standard manual versus automated RCM U-Net) (p=0.7). The cortico-medullary area ratio measured by the RCM U-Net was significantly increased in case of cellular rejection by comparison to humoral rejection (1.6 +/- 0.39 versus 0.99 +/- 0.32, p=0.019)

Reviewer: 2

We thankful to reviewer 2 comments and feedback, which helped us to improve the paper's clarity and concision. The response to the Reviewer 2 comments are given below:

1. There are a huge number of acronyms. I appreciate the challenges with this work but at no point could I find u-NET described and as a clinician this was very confusing. please describe what this is. Likewise, many other values are presented as a series of letters which limited the understanding of this paper.

Answer: A deep learning architecture includes multiple convolutional layers followed by activation layer, pooling layer, dropout, and batch normalization layer. U-Net is one of the popular used neural networks in deep learning. U-Net is an encoder-decoder network consists of multiple convolutional layers followed by activation layer, pooling layer, dropout, and batch normalization independent from encoding and decoding workflows (added in the revised manuscript Line: 91-94). The U-Net architecture looks like a ‘U’ shape which justifies its name. 

For other names description a List of acronyms is given as supplementary file.

2. Please supply some clinical information about how patients recruited etc.

Answer: This study included adult kidney transplant recipients and CKD patients who were scheduled for a kidney biopsy for clinical grounds. MRI was performed on the same day as the biopsy wherever possible, or within one week. Patients over the age of 18 who were being monitored at our hospital were eligible for enrolment. Pregnancy, claustrophobia, and patient refusal were all exclusion criteria. Additional fasting serum and urine were collected and kept at -80°C in all individuals. The Table 1 presents the baseline characteristics of the study population (n=114): clinical parameters, medication, laboratory measurements, biopsy diagnosis and chronic histological lesions at the time of inclusion. (added in revised manuscript Line: 215-231).

A table is added in the revised manuscript as Table 1 and order of tables numbers are updated accordingly.

3. Some baseline demographics would be helpful.

Answer: Given in Table 1 in the revised manuscript.

4. What eGFR formula was used?

Answer: The eGFR was calculated according to the CKD-EPI equation (added in manuscript line 125-126)

5. I could not find a figure legend- maybe it was lost in formatting.

Answer: Updated in the revised manuscript

6. Typo in figure 3.

Answer: Updated in the revised manuscript

---

## [Decision Letter · Decision Letter 1]

25 Oct 2022

Validation of automatically measured T1 map cortico-medullary difference (ΔT1) for eGFR and Fibrosis assessment in allograft kidneys

PONE-D-22-08551R1

Dear Dr. Vallée,

We’re pleased to inform you that your manuscript has been judged scientifically suitable for publication and will be formally accepted for publication once it meets all outstanding technical requirements.

Kind regards,

Alfredo Vellido

Academic Editor

PLOS ONE

Additional Editor Comments (optional):

Reviewers' comments:

Reviewer's Responses to Questions

**Comments to the Author**

1. If the authors have adequately addressed your comments raised in a previous round of review and you feel that this manuscript is now acceptable for publication, you may indicate that here to bypass the “Comments to the Author” section, enter your conflict of interest statement in the “Confidential to Editor” section, and submit your "Accept" recommendation.

Reviewer #1: (No Response)

Reviewer #2: All comments have been addressed

2. Is the manuscript technically sound, and do the data support the conclusions?

Reviewer #1: Yes

Reviewer #2: Yes

3. Has the statistical analysis been performed appropriately and rigorously? 

Reviewer #1: Yes

Reviewer #2: Yes

4. Have the authors made all data underlying the findings in their manuscript fully available?

Reviewer #1: Yes

Reviewer #2: Yes

5. Is the manuscript presented in an intelligible fashion and written in standard English?

Reviewer #1: Yes

Reviewer #2: Yes

6. Review Comments to the Author

Reviewer #1: Dear Author/s

Greetings for you

I appreciate your effort and work

It is appropriate to publish the article with the current changes. Thank you for sharing your work with us.

Kind regards

Reviewer #2: Thank you for addressing my comments and those of the other reviewer. I have no further comments to make

7. PLOS authors have the option to publish the peer review history of their article (what does this mean?). If published, this will include your full peer review and any attached files.

Reviewer #1: **Yes: **Yavuz AYAR

Reviewer #2: No

---

## [Editor Report · Acceptance letter]

2 Nov 2022

PONE-D-22-08551R1 

Validation of automatically measured T1 map cortico-medullary difference (ΔT1) for eGFR and Fibrosis assessment in allograft kidneys 

Dear Dr. Vallée:

I'm pleased to inform you that your manuscript has been deemed suitable for publication in PLOS ONE. Congratulations! Your manuscript is now with our production department. 

Kind regards, 

on behalf of

Dr. Alfredo Vellido 

Academic Editor

PLOS ONE